# An Integrated Deep Learning Framework Enables Rapid Spatiotemporal Morphodynamic Predictions Toward Long-Term Simulations

- Mohamed M. Fathi<sup>1,2</sup>, Zihan Liu<sup>3</sup>, Anjali M. Fernandes<sup>4</sup>, Michael T. Hren<sup>5</sup>, Dennis O. Terry, Jr.<sup>6</sup>, C. Nataraj<sup>3</sup>, and Virginia Smith<sup>7</sup>
  - <sup>1</sup> Dept. of Civil Engineering, Florida Gulf Coast University, United States;
- <sup>2</sup> Dept. of Civil Engineering, Faculty of Engineering, Fayoum University, Egypt;
  - <sup>3</sup> Villanova Center for Analytics of Dynamic Systems and Department of Mechanical Engineering, Villanova University, United States;
  - <sup>4</sup> Dept. of Earth and Environmental Sciences, Denison University, United States;
  - <sup>5</sup> Dept. of Earth Sciences, University of Connecticut, United States;
- 6 Dept. of Earth and Environmental Science, Temple University, United States;
  - <sup>7</sup> Dept. of Civil and Environmental Engineering, Villanova University, United States.

Correspondence to: Mohamed M. Fathi (m.fathi.said0@gmail.com & msaid@fgcu.edu)

https://doi.org/10.5194/egusphere-2025-3368 Preprint. Discussion started: 24 November 2025 © Author(s) 2025. CC BY 4.0 License.

## Abstract

Physics-based morphodynamic modeling is essential for advancing river management science and understanding Earth's geomorphological evolution processes. However, their computational demands and long processing times hinder long-term applications. This paper introduces and tests a robust Deep Learning (DL) framework that opens the door to overcoming these challenges through integrating convolutional neural networks (CNNs) with long short-term memory algorithms (LSTM). This advancement facilitates rapid and continuous spatiotemporal predictions of hydrodynamic parameters and morphodynamic responses of flood events. Hydrodynamic predictions showed strong performance across the testing dataset, with mean RMSEs of 0.15 m and 0.04 m/s for water depth and flow velocity, respectively. Bed change predictions also demonstrated promising results, with normalized RMSE of 27% and R2 of 0.93. This novel approach generates predictions 4700 times faster than traditional physics-based computational models, representing a paradigm shift in long-term river evolution simulations and pioneering new frontiers in fluvial morphodynamic modeling.

35

40

25

30

#### Summary

Understanding and predicting the evolution of river landscapes is critical for effective river management. Traditional physics-based morphodynamic models, while accurate, are computationally intensive and often impractical for long-term applications. This study presents a robust deep learning framework, which was designed to overcome the computational limitations by enabling rapid and reliable predictions of hydrodynamic and sediment transport behaviors.






# 1 Introduction

Fluvial landscapes are the nexus of the water cycle, climate, and Earth surface processes. Easy access to water for domestic and agricultural needs has made river floodplains attractive places to live throughout human history (Fang and Jawitz, 2019). Floodplains are morphodynamically active environments continuously modified by erosional and depositional processes during floods, coupled with lateral migration of adjacent river channels. These changes can be accompanied by significant ecological and socio-economic impacts, including migration or destruction of sensitive habitats, loss of agricultural lands, changes in navigability, destruction of infrastructure, and loss of life (Mananoma, 2009; Vercruysse et al., 2017; Zakipour et al., 2023). Associated risks can only be offset through effective long-term (multi-century) management planning for reach-scale geomorphic responses to environmental perturbation (Fathi et al., 2024). This is especially important as we plan for the impacts of climate change on rivers, floodplains, and the communities they support.

Sediment transport and morphodynamic responses of rivers and floodplains have been recorded and/or studied using remotely sensed data and projected using physics-based numerical models (Donati et al., 2021). While significantly advancing sediment transport and morphodynamic theory, each approach is limited. Remote sensing techniques, utilizing satellites, airplanes, and remote vehicles (e.g., drones), represent the most practical and precise approach for morphodynamic mapping (Boothroyd et al., 2021). They are limited by coarse spatial resolution in freely available imagery, high costs for high-resolution alternatives, and short decadal record lengths (Grabowski et al., 2014). Further, while extremely useful in understanding past landscape changes, these techniques are relatively limited in their applicability to future change projections in assorted synthetic scenarios, e.g., climate change studies and restoration assessment projects. Numerical models, on the other hand, can be used to understand and project past and future landscape change, respectively as they include options for synthetically modeling environments with different climatic conditions, sediment properties, and/or channel geometries (Coulthard and Van De Wiel, 2012). The research community efforts have developed and advanced multiple physics-based geomorphological models, such as Delft3D (Deltares, 2010), MIKE 21C (DHI, 2017), and HEC-RAS (USACE, 2021). However, these are computationally expensive models, leaving users with no option but to compromise on accuracy and/or the time scales to which they are applied (Gonzales-Inca et al., 2022).

The computer revolution brought the widespread usage of geomorphologic models to simulate hydrodynamic and sediment transport in rivers. Models incorporate various levels of complexity, ranging from the simplest 1D models to the most complex/advanced 3D modeling (Williams et al., 2016). Choosing among these modeling dimensions requires a trade-off between accuracy, computational resources, and applicability. The 2D approach offers a balanced combination of accuracy







and computational efficiency, enabling investigations across multiple morphological applications, e.g., river restoration (McDonald et al., 2016), the influence of dam presence (Giri et al., 2019) and dam removal (Gelfenbaum et al., 2015) on river morphology, eco-hydrology of fish habitats (Wheaton et al., 2018), vegetation dynamics (Best et al., 2018), bedform simulations (Chen et al., 2012), and bar morphology (Kasprak et al., 2019). Despite these wide-ranging applications, the 2D modeling approach is hampered by relatively long processing time and limitations on spatial and temporal scales of applicability, e.g., morphological change on relatively short between 10<sup>0</sup>-10<sup>2</sup> km river reaches (Brasington and Richards, 2007) and timescales of just weeks or months (Williams et al., 2016). This is primarily due to: a) a need for small computational time-steps to ensure model stability (De Goede, 2020), and b) limited usage of parallel computing and/or limited capability to incorporate high computing power capabilities, e.g., graphics processing units (Karim et al., 2023).

To develop more efficient techniques for both flood hydrodynamic and geomorphological modeling, the scientific community sought solutions in Machine Learning (ML) approaches (Karim et al., 2023). Recent achievements include the usage of ML algorithms to predict flood extent maps (Avand et al., 2022; Bentivoglio et al., 2022; Ma et al., 2021; Madhuri et al., 2021; Mehedi et al., 2022; Talukdar et al., 2021). Advances in flood extent mapping spurred the development of more advanced models to predict water depth maps of flood events, e.g., artificial neural networks (ANNs) and Convolutional Neural Networks (CNNs) (Chu et al., 2020; Kabir et al., 2020). In the field of sediment transport, researchers have employed various algorithms, including random forest and support vector regression (Kwon et al., 2022), Long Short-Term Memory (LSTM) (Kaveh et al., 2021), M5 model tree (Ouellet-Proulx et al., 2016), and gated recurrent units (Huang et al., 2021), to predict suspended sediment concentrations. The focus has further extended to explore the prediction of specific sediment transport characteristics such as bed load material in channels (Hosseiny et al., 2023; Kitsikoudis et al., 2014; Sahraei et al., 2018), sediment yield during monsoon (Ghose, 2018), and sediment transport dynamics within sewer systems (Zounemat-Kermani et al., 2020). Despite these scientific efforts to advance sediment load simulations, to the best of our knowledge, no prior research has explored the usage of data-driven approaches in predicting the 2D geomorphological responses within floodplains.

To address this gap, this study aims to leverage the capabilities of Deep Learning (DL) approaches to efficiently simulate the hydrodynamic and morphodynamic behavior within floodplains. Most recently, Fathi et al. (2025) developed a Hybrid DL framework for Flood Mapping (HDL-FM) to simulate the 2D flood dynamic characteristics. This framework integrates the spatial advantages of CNN along the sequential capabilities of LSTM, to predict three essential hydrodynamic features: water depth, flow velocity, and flow direction maps. The HDL-FM framework demonstrated efficiency and robustness in capturing the spatiotemporal flood dynamic nature. Here we extend the HDL-FM framework to consider not only the hydrodynamic characteristics but also the geomorphologic behavior

https://doi.org/10.5194/egusphere-2025-3368 Preprint. Discussion started: 24 November 2025 © Author(s) 2025. CC BY 4.0 License.







mapping. The primary objective is to develop an efficient framework capable of capturing the complexity of morphodynamic processes while avoiding the computational challenges posed by traditional approaches. The resulting framework predicts the dynamics of essential hydrodynamic outputs: water depth and flow velocity, which are fundamental inputs to the morphodynamic target, represented in bed change maps. This approach presents a robust and more efficient methodology that has significant implications in several floodplain engineering applications, including 2D sediment transport simulations, assessing the impacts of climate change on the rivers' geomorphology, river rehabilitation projects, and, notably, in the realm of long-term land evolution studies.

## 2 Materials and Methods

## 2.1 Study Area and Geomorphologic Model

This study focuses on a 22 km segment of the Ninnescah River in central Kansas, a tributary of the Arkansas River (Figure 1-a). The Ninnescah River primarily has sandy banks, resulting in typically wide, shallow, and straight channels, with a bankfull width of approximately 100 m (Costigan et al., 2014). The selected river segment, with a sinuosity coefficient of two, provides an ideal case study for this research. The hourly discharge data of this segment is monitored by the United States Geological Survey (USGS) station 07145500, with a mean annual discharge of approximately 15 cms.

A 2D geomorphological simulation for the Ninnescah River segment was generated using HEC-RAS. The developed model covers a 2D grid area of 66.5 km2 with 520 x 320 cells (each 20 m by 20 m). Three essential inputs are required for this type of simulation: a Digital Elevation Model (DEM) for the river including the floodplain with a high spatial resolution of 1/3 arc-second (approximately 10 meters) from USGS (2018), a flow hydrograph at the upstream boundary (USGS gage 07145500), and riverbed sediment characteristics based on the sediment samples of the riverbed collected by Costigan et al. (2014). Sediment grain sizes at D10, D50, and D90 measured 0.25 mm, 0.45 mm, and 1.1 mm, respectively. Due to the absence of bathymetry information in the DEM file, a trapezoidal cross-section profile was burned into the DEM file (Choné et al., 2018; Costigan et al., 2014). Lastly, the hydrograph component represents the driving hydrodynamic power of the flow through the system, from USGS at station 07145500.

A hydrograph with a broad range of flood events is essential to enhance the capabilities of the DL framework in capturing hidden patterns in the mapping between inputs and outputs. Unfortunately, generating sufficiently long training and testing datasets is hindered by the long processing time using the physics-based model. Additionally, significant portions of the observed hydrograph are baseflow, which corresponds to a period of relatively low flow with minimal sediment transport. To address these challenges, a constructed hydrograph of 20 events was extracted from the observed hydrograph to avoid

long periods of baseflow discharge. This hydrograph was split into three portions: training, validation, and testing of 11, 3, and 6 events, respectively (Figure 1-b).

Figure 1: The case application: a) the location of the Ninnescah River segment, Kansas, USA, and b) the hourly hydrograph, divided into training, validation, and testing sets for the DL modeling.

The Morphological Acceleration Factor (M<sub>t</sub>) technique is used in the HEC-RAS model to reduce the sediment transport simulation time.  $M_f$  is a scalar quantity that is used to reduce the timestep values of a hydrograph by  $M_6$  while multiplying the calculated erosion and deposition rates by the same factor (Lesser et al., 2004). It is worth noting that there is a trade-off between utilizing a higher acceleration factor and the accuracy of the morphodynamic outputs. Morgan et al. (2020) explored this trade-off on Nooksack River in Washington through investigating a range of M<sub>f</sub> from 5 to 50. The results indicated that the lowest  $M_f$  of 5 reduced the processing time by 80%, with a relatively low absolute percentage error of 8%. Consequently, each flood event in our study spans a full-scale of 10-







days, encompassing the typical progression of flood events from initial baseflow, through the peak, and returning to baseflow, which is reduced to only 2 days using  $M_f$  of 5 (Figure 1-b).

## 160 2.2 Deep Learning Model

This study aims to extend the capabilities of the Hybrid DL framework for Flood Mapping (HDL-FM), developed by Fathi et al. (2025), into geomorphology mapping (HDL-GM). This hybrid DL framework integrates the spatial strengths of Convolutional Neural Networks (CNN) coupled with the temporal prowess of Long Short-Term Memory (LSTM). CNNs perform automated feature engineering through learnable filters, enabling the detection and extraction of spatial features from the inputs. Multiple layers of CNNs build a hierarchy of increasingly complex features, ultimately enabling the recognition of objects within the 2D grids (Bhatt et al., 2021). LSTM is a powerful type of recurrent neural network designed to handle sequential data (Yu et al., 2019). LSTMs incorporate memory cells with input, output, and forget gates, allowing the network to selectively remember or forget information over long sequences.

HDL-FM is an integrated DL model combining CNN and LSTM (Figure 2-a), originally developed to predict the hydrodynamic properties of flood events in 2D grids. This study incorporates this approach into geomorphologic modeling by using it to capture sediment transport. This framework encompasses three models, all of which have the same architecture (Figure 2-a), but with different targets; water depth, flow velocity, and bed change. This framework utilizes a uniform 2D shape of 520 x 320 grid (each 20 m by 20 m) for both inputs and targets, ensuring compatibility with the HEC-RAS model simulation. Each model commences with an encoder block comprised of three CNN layers, each followed by a rectified linear unit (ReLU) activation function. A 2x2 max-pooling operation was applied to each layer, balancing the preservation of high-resolution morphology features in narrow river segments with improved computational efficiency through a reduction of grid size by half. It should be noted that large pool sizes tend to lose important details in the segment meanders, affecting morphodynamic predictions. In the encoder phase, a flattening process is applied, generating a single vector suitable for processing by a single LSTM layer. The enhancement in prediction accuracy through utilizing additional LSTM layers was minimal. An ANN dense layer is introduced as a decoder stage where the LSTM outputs convert back into a long vector possessing the same number of elements as the physics-based grid, to reconstruct the original grid dimension.

This study explores the applicability of a hybrid DL model to predict the hydrodynamic and morphodynamic behavior of flood events at 30-min intervals according to three models: water depth, flow velocity, and bed change. Each model utilizes a distinct set of inputs as described in Figure 2-b. In contrast to the original framework, the geomorphological version, HDL-GM, incorporates a temporally varying topographic input (*Z*). To standardize the topography input, a simple normalization


process is applied through subtracting the average elevation value, calculated at the beginning of the simulation, from the entire topography grid. Additionally, other dynamic inputs are introduced to the framework, including water depth (D), flow velocity (V), and discharge value (Q) at the upstream boundary condition. The upstream discharge series is transformed into a grid initialized with zeros, where the Q value is subsequently assigned to a 7x7 cell block at the upstream point of the reach. The proposed three models employ lagged input variables from the previous time-step to predict the targets at the subsequent time-step targets, such as the topography and water depth grids (Figure 2-b). Conversely, other inputs, encompassing the flow velocity and upstream discharge, are utilized from the current time-step.

Figure 2: An overview of the HDL-FM model, where a) model architecture, b) model inputs and targets, and c) sequential model integration for the testing procedure.

The training process for the proposed framework employs the smooth  $L_1$  loss function for enhanced convergence stability and robustness to outliers (Girshick, 2015). Optimization is performed using the Adam optimizer (Kingma and Ba, 2015), with the learning rate dynamically adjusted






throughout the training process using the ReduceLROnPlateau technique to ensure efficient and rapid convergence (Al-Kababji et al., 2022; PyTorch, 2024).

#### 210 2.3 Model Evaluation and Performance Criteria

Testing Mechanism:

The dynamic nature of geomorphic simulations, which continuously evolves over time, depends on lagged inputs from previous time-steps to predict the system's behavior at the following time-steps. Evaluating the capabilities of these models to operate independently, without prior reference inputs, is essential. To address this, a four-step loop was developed to assess the accuracy of the proposed framework in real-world applications (Figure 2-c). At a typical time-step, the framework starts with predicting the water depth, which serves as an essential input for the subsequent step of predicting the flow velocity. In the third step, the bed change of the topography is predicted based on outputs from the first two steps. Finally, the loop concludes by using the bed change output to update the topography grid, which serves as the primary input of the loop at the next time-step. This loop is applied iteratively to the testing dataset, allowing the framework to predict its own future inputs without relying on any reference inputs. This testing technique is introduced to evaluate the framework's robustness for long-term simulations, to ensure that it can operate without error accumulation issues.

#### Performance Criteria:

To evaluate the accuracy of the proposed model in predicting the hydrodynamic and bed change variables, Root Mean Square Error (RMSE) is used to quantify the average magnitude of errors between simulated and reference datasets (Eq. 1) (Stigler, 1990), and assesses the accuracy at a single time-step or is averaged across the testing period. However, using the total number of cells (including the cells without any morphodynamic behavior) in estimating the average could lead to misleading accuracy values. To address this challenge, the cells considered in the RMSE are restricted to the active cells, which represent the locations within the predicted or reference grids where water depth or bed change exceeds 0.05 m or 0.02 m, respectively. However, because the bed change values increase progressively over time, RMSE alone can be misleading to track the model performance over time. To address this, Normalized Root Mean Square Error (NRMSE) is introduced to normalize RMSE into a percentage relative to the root mean square of the target values (Eq. 2), thereby facilitating a more meaningful and consistent comparison of prediction errors across different time steps (Mentaschi et al., 2013). Additionally, the coefficient of determination  $(R^2)$  is employed as a complementary metric (Eq. 3) (Veall and Zimmermann, 1996). R<sup>2</sup> evaluates how effectively the model captures the underlying distribution of the data, in this case regarding erosional and depositional activities across the grid domain.





RMSE = 
$$\sqrt{\frac{1}{N} \sum_{i=1}^{N} (R_i - S_i)^2}$$
 Eq. 1

NRMSE = 
$$\frac{\text{RMSE}}{\sqrt{\frac{1}{N}\sum_{i=1}^{N}(R_i)^2}}$$
Eq. 2

$$R^{2} = 1 - \frac{\sum_{i=1}^{N} (R_{i} - S_{i})^{2}}{\sum_{i=1}^{N} (R_{i} - \bar{R})^{2}}$$
 Eq. 3

where  $R_i$ ,  $S_i$  are the reference and the simulated values, respectively, and  $\bar{R}$  is the average value of reference grids. N is the number of active cells within the predicted or reference domains.

#### 3 Results and Discussion

Two distinct versions of the HDL-GM framework were tested. These versions are based on the nature of the training dataset: Event-Based (EB) and Continuous-Based (CB) datasets (Figure 1-b). The EB dataset aggregates discrete events that are simulated independently using the physics-based model, thereby enhancing computational efficiency in generating the training dataset by allowing parallel simulations. Conversely, the CB dataset demands significantly longer processing time due to its continuous simulation of a series of events. To assess the performance and trade-offs of these models, three experimental scenarios were implemented: 1) the EB-trained framework tested on the EB dataset, 2) the EB-trained framework tested on the CB dataset, and 3) the CB-trained framework tested on the CB dataset. These scenarios were designed to investigate the tradeoff between the computational efficiency of dataset generation and the accuracy of the framework in both EB and CB applications.

# 3.1 EB-Trained HDL-GM Framework for Geomorphic Simulation

For the EB-trained HDL-GM framework, both water depth and flow velocity models exhibit strong agreement with reference data from HEC-RAS, where the average RMSE values, across the entire testing dataset, are 0.19 m and 0.04 m/s, respectively. These low RMSE values not only underscore the robust hydrodynamic capabilities of this approach but are also essential for accurate geomorphic predictions.

The HDL-GM framework integrated the outputs of both hydrodynamic models, water depth and flow velocity, as inputs for the bed change model. Figure 3 illustrates the bed change results at the end of three testing events. The bed change model within the proposed framework exhibited robust capabilities in accurately predicting the spatial patterning of erosional and depositional processes across the grid domain, as compared to HEC-RAS results. This is evidenced by the low RMSE values, across the testing six events, ranging from 0.02 to 0.03 m, with a mean of 0.026 m. It should be noted that most of the high relative errors were concentrated in shallow wetland areas, where the morphodynamic

activities are primarily forced by shallow overbank flow. This may be attributed to the minimal magnitude of bed changes in these regions, with high uncertainties. Consequently, when the error is normalized by these small bed changes, it results in high relative errors.

Figure 3: Comparison of bed change predictions of EB-testing dataset by HEC-RAS and EB-trained HDL-GM framework for the three testing events, illustrating the spatial distribution of absolute error and relative error between both models, where Ts denotes a specific flood event in the testing dataset.



The EB-trained framework exhibited robust performance in accurately predicting the bed change of the EB-testing dataset. However, it is crucial to assess its performance on the CB-testing dataset, which is characterized by cumulative geomorphic behavior across multiple events. Figure 4 presents the absolute and relative error results at the end of multiple testing events within the CB dataset. The EB-trained framework maintained acceptable performance during the first couple of events; but after three events, a noticeable accumulation of error values emerged. By the end of the CB simulation, the relative error exceeded 100% at numerous locations both within the stream channel and shallow areas. While the EB-trained HDL-GM framework, based on an efficient EB-training dataset, demonstrated strong predictive capabilities for the EB-testing dataset, its performance on the CB-testing dataset revealed significant and unacceptable error accumulation. This behavior could potentially be attributed to the nonlinearity of sediment transport over multiple flood events, which may not be fully represented in the EB dataset. These findings underscore its limitations in capturing the complexities of continuous morphodynamic processes.

Figure 4: Absolute error and relative error of using EB-trained HDL-GM framework to predict bed change for CB-testing dataset.

# 3.2 CB-Trained HDL-GM Framework for Geomorphic Simulation

The CB-trained framework exhibited strong performance in predicting hydrodynamic features, with mean RMSE values of 0.15 m for water depth and 0.04 m/s for flow velocity variables, across the entire testing dataset. These hydrodynamic outputs were subsequently integrated into the bed change model. Figure 5 presents the absolute and relative error results at the end of multiple testing events within the CB dataset. Initially, the CB-trained framework maintained relatively high relative




errors at the end of the first testing event. However, it exhibited rapid recovery, yielding substantially reduced relative errors from the second testing event. This robust performance continued throughout the CB testing dataset, with no accumulation of relative errors. This is further evidenced by an RMSE value of 0.07 m at the end of the sixth event. These findings highlight the capabilities of the CB-trained framework to predict morphodynamic behavior across a testing dataset encompassing a series of continuous events.

Figure 5: Absolute error and relative error of testing CB-trained HDL-GM framework to predict bed change for CB-testing dataset.

# 3.3 Statistical Comparison of EB and CB-Trained Frameworks

To evaluate the performance of the EB and CB-trained frameworks, a comprehensive statistical comparison was conducted across three previously defined scenarios. The assessment utilized four performance criteria: RMSE, NRMSE, 95% error, and R², as represented in Figure 6. The 95% error criterion indicates that 95% of the active cells exhibit an error below this value. For the EB-testing dataset, results are presented separately at the end of each event. In contrast, the results for the CB-testing dataset are cumulative, extending from the start of the simulation to the end of a given event. The key findings from this comparative analysis are outlined as follows.

 Scenario 1: the EB-trained framework demonstrated robust performance when applied to the EB-testing dataset. Performance metrics indicated consistent predictive accuracy with mean NRMSE and R<sup>2</sup> of 22% and 0.94, respectively; this suggests that the framework effectively captures event-based dynamics.




- Scenario 2: when applied to the CB-testing dataset, the EB-trained framework initially showed acceptable performance for the first two events. However, subsequent events revealed a significant degradation in predictive accuracy due to error accumulation. The NRMSE escalated from an initial value of 21% to 113% by the end of the sixth event, which yielded a negative R<sup>2</sup> value. Over time, the error becomes untenable.
- Scenario 3: the CB-trained framework exhibited superior adaptability and stability when applied to the CB-testing dataset. The framework's performance improved markedly after the first one or two events. The NRMSE decreased from an initial value of 66% to less than 27%, at the end of the testing dataset. It is worth noting that the RMSE values demonstrated remarkable stability, within a narrow band of 0.06 to 0.07 m, throughout the simulation.

Figure 6: Evaluation statistics of the EB and CB-trained frameworks on the EB and CB-testing datasets.

## 3.4 The Influence of Temporal Resolution on CB-Trained HDL-GM Accuracy

The temporal resolution of a model framework profoundly influences training efficiency, prediction accuracy, and the model's capacity to fulfill specific application requirements, thereby shaping its overall effectiveness and utility. The CB-trained HDL-GM framework was assessed using datasets with varying time-steps, ranging from 0.25-hr to 16-hr (Figure 7). The framework trained on datasets with time-steps of 0.25, 0.5, and 1-hr time-steps demonstrated relatively consistent performance, maintaining a consistent NRMSE of less than 30%. However, coarser time-steps, of 2, 4, and 8-hr, led to a decline in predictive capabilities, as evidenced by a reduction in R<sup>2</sup> values from 0.90 to 0.76. This performance deteriorated significantly for the 16-hr time-step configuration, exhibiting an NRMSE exceeding 100% and a negative R<sup>2</sup> value. This analysis suggests that beyond the 8-hr threshold, coarser resolution reduces the model's ability to accurately capture the hydrodynamic peaks of flood events, thereby impacting the accuracy of morphodynamic predictions. These findings suggest that there is a potential relationship between the hydrodynamic scale of the system and the optimal temporal resolution of the modeling framework, warranting further research in this area as a future endeavor.





Figure 7: Evaluation statistics of the CB-Model for various temporal resolutions at the end of the simulation.

## 3.5 Comparison of Prediction Runtimes Between Physics-Based and HDL-GM Models

The prediction runtime and computational demands have been critical considerations in the morphodynamic modeling approach. A comprehensive processing time assessment was conducted by comparing the HEC-RAS model, executed on an Intel-based laptop, with the HDL-GM framework, trained on both the same laptop system and Augie, Villanova Engineering's High-Performance Computing Cluster (HPC), as presented in Table 1. Building upon the previous section's findings, the HDL-GM framework was trained using a dataset with a 1-hr time-step, due to its proven efficiency and accuracy. The results of the training stage and the predictions of a 528-hr duration are presented in Table 1. The key findings from this comparative analysis are outlined as follows.

- The processing time of HEC-RAS is considerably long, requiring over 13 days to complete a 528-hr simulation.
  - The HDL-GM framework's training stage exhibited significant efficiency, particularly in the HPC environment, where parallel training of the three models was completed in about 2.25-hr.
     In contrast, training on a standard laptop, which necessitated a series approach, required about 11-hr.
  - The prediction runtime for the trained framework demonstrated remarkable efficiency. By
    executing the loop between the water depth, flow velocity, and bed change models, the HDLGM framework operates at a rate that is 4700 times faster than the physics-based model,
    regardless of whether the HPC or the laptop was utilized.

Table 1: Computational time of HEC-RAS vs HDL-GM at two distinct computational systems. All time values are in minutes

| varies are in influees. |         |        |          |             |  |  |  |  |  |
|-------------------------|---------|--------|----------|-------------|--|--|--|--|--|
| Model                   | Machine | Target | Training | Predictions |  |  |  |  |  |

|         |        |                    | Dragoning time  | Total        | Total |
|---------|--------|--------------------|-----------------|--------------|-------|
|         |        |                    | Processing time | time         | time  |
| HEC-RAS | Laptop |                    |                 |              | 18900 |
|         | Laptop | Water depth        | 213             |              |       |
|         |        | Velocity magnitude | 142             | 671 (series) | 4     |
| IIDI CM |        | Bed change         | 316             |              |       |
| HDL-GM  | HPC    | Water depth        | 136             | 126          |       |
|         |        | Velocity magnitude | 73              | 136          | 4     |
|         |        | Bed change         | 112             | (parallel)   |       |

Laptop: Intel® Core<sup>TM</sup> i9-13900H + 32 GB LPDDR5 RAM

HPC: 64 AMD EPYC Series CPU Cores

## 4 Conclusions






Computational morphodynamic models are important for simulating the erosional and depositional processes associated with moving rivers and changing landscapes. However, the computational demands of these models result in prolonged processing times, thereby limiting their utility in long-term simulations. To address this shortcoming, many researchers have explored data-driven algorithms in predicting sediment transport rates with numerous applications. However, to our knowledge, no prior research has focused on leveraging the capabilities of DL in predicting 2D geomorphic evolution of river floodplains.

This study introduces a novel framework that harnesses DL capabilities to address the computational challenges inherent in physics-based models, yielding robust, rapid, and reliable 2D morphodynamic maps. This approach represents a major step in geomorphologic prediction, through a full-system simulator of the complex interaction of hydrodynamics of flood events and morphodynamic processes in fluvial areas, through integrating the spatial advantages of CNN with temporal sequences of the LSTM algorithm. The effectiveness of this spatiotemporal framework represents a powerful extension of most DL implementations, which are either sequentially dynamic point-scale applications or static spatially distributed simulations (Bennett et al., 2024). The testing technique was developed to assess the framework's capabilities in real-world scenarios, where the three models, water depth, flow velocity, and bed change, operate iteratively in a loop.

Additionally, a comprehensive analysis was conducted to evaluate the trade-off between the efficiency of generating training datasets and their corresponding predictive accuracy. Two methodologies were compared: EB, representing an efficient dataset generation approach, and CB, a computationally intensive alternative, through three experimental scenarios: 1) the EB-trained framework tested on the EB dataset, 2) the EB-trained framework tested on the CB dataset, and 3) the CB-trained framework tested on the CB dataset. Some key findings from this study include the following.

- Both EB and CB-trained frameworks exhibited strong performance in predicting both hydrodynamic features: water depth and flow velocity, whereas the CB-trained framework had slightly superior performance.
- Despite the robust capabilities of the EB-trained framework in predicting the spatiotemporal bed change activities of the EB-testing dataset, its performance on the CB-testing dataset exhibited significant limitations, characterized by a pronounced accumulation of errors. As a result, the model performance significantly decreased after a few flood events.
- The CB-trained framework demonstrated superior performance and strong stability when
   applied to the CB-testing dataset, with a very narrow range of RMSE of 0.06 to 0.07 m,
   throughout the testing evaluation, showing consistent performance over time.
  - Consistent accuracy in simulating morphodynamics was achieved across multiple temporal resolutions: 0.25, 0.5, and 1-hr, demonstrating the framework's versatility in meeting diverse application requirements. However, as the timesteps became coarser, the model performance became worse.
  - The HDL-GM framework achieved a remarkable speedup of over 4700 compared to the HEC-RAS model. This significant efficiency gain is secured through the transition from a secondbased time-step of physics-based models to an hourly time-step, by leveraging the capabilities of the DL approach.
- Ultimately, the proposed framework demonstrates significant potential in enhancing morphodynamic modeling in fluvial rivers and floodplains, particularly from the long-term perspective, climate change assessments, river rehabilitation projects, and other investigations requiring long-term simulations. Despite these promising capabilities, its primary limitation lies in its inability to effectively simulate unseen topographical conditions. Additionally, the current implementation of the HDL-GM framework is for a particular set of hydrodynamic and sediment transport properties, especially Manning n, grain-size distributions, and cohesive properties. To address these limitations, future work is needed to improve the generalizability and performance by integrating essential hydrodynamic and sediment transport physics into the DL framework (Karniadakis et al., 2021; Mohamad et al., 2021).

# 425 Acknowledgments

This work was supported by NSF under Award No: 1844180. For this study, the authors utilized the Augie High-Performance Computing cluster, at Villanova University. The authors would

also like to acknowledge and thank Dr. Aaron Wemhoff for his assistance and for sharing his insight with Augie. This work was also partially supported by the Villanova Center for Resilient Water Systems (VCRWS). Nataraj and Liu were partially supported by a grant from Office of Naval Research (N00014-22-1-2480, PM: Lynn Petersen).

# **CRediT** authorship contribution statement

Mohamed M. Fathi: Conceptualization, Methodology, Software, Validation, Data Curation, Writing - Original Draft, Visualization. Zihan Liu: Methodology, Funding acquisition, Validation, Writing - Review & Editing. Anjali M. Fernandes: Conceptualization, Funding acquisition, Writing - Review & Editing. Michael T. Hren: Conceptualization, Funding acquisition, Writing - Review & Editing. Dennis O. Terry: Conceptualization, Funding acquisition, Writing - Review & Editing. C. Nataraj: Conceptualization, Methodology, Funding acquisition, Validation, Writing - Review & Editing. Virginia Smith: Conceptualization, Methodology, Supervision, Funding acquisition, Validation, Writing - Review & Editing.

# **Open Research Section**

The codes and training dataset that were used to build the HDL-GM model to simulate the flood and geomorphic dynamic maps using Python language, based on Pytorch library, can be found on GitHub: <a href="https://github.com/m-fathi-said/Flood-Modeling-HDL-FM.git">https://github.com/m-fathi-said/Flood-Modeling-HDL-FM.git</a>. This repository was created by Mohamed M. Fathi (<a href="mailto:msaid@fgcu.edu">msaid@fgcu.edu</a> / <a href="mailto:msaid@gmail.com">m.fathi.said@gmail.com</a>) in 2024 under the MIT License.

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
