# Peer review of "An Integrated Deep Learning Framework Enables Rapid Spatiotemporal Morphodynamic Predictions Toward Long-Term Simulations"

_EGUsphere, 2025_

## Referee Comment (RC1)

I was invited to review the manuscript by Mohamed et al. (2025), which presents an integrated deep learning (DL) framework trained on outputs from a physics-based morphodynamic model to efficiently accelerate spatiotemporal predictions of river–floodplain evolution. While the framework does not aim to introduce new physical concepts or process understanding, it effectively leverages deep learning to reproduce the behavior of the reference physics-based model in a computationally efficient manner. Consequently, the primary contribution of the manuscript lies in methodological advancement and workflow efficiency rather than in the development of new morphodynamic theory. Overall, the study is interesting and promising. Nevertheless, several points would benefit from clarification or further refinement. Addressing these comments to the satisfaction of the editor and reviewers would, in my view, strengthen the manuscript, and I would be happy to recommend it for publication.

1. The main focus of this study is to demonstrate that deep learning (DL), trained on outputs from the physics-based model, can reproduce comparable results at a substantially higher computational speed. It does not explore the broader potential of DL for simulating hydrodynamic and morphodynamic processes in river or floodplain systems. Therefore, the main contribution lies in computational efficiency rather than providing new physical insights. In addition, numerous studies have applied deep learning to achieve higher computational speeds; for example, Synthetic in Bentivoglio et al. (2022, *Deep learning methods for flood mapping: a review of existing applications and future research directions*) and Karim et al. (2023, *A review of hydrodynamic and machine learning approaches for flood inundation modeling*). Consequently, the statement in the abstract (line 34) claiming that this study represents "pioneering new frontiers in fluvial morphodynamic modelling" is misleading and should be revised.

2. The abstract reports good performance metrics (e.g., RMSE, $R^2$), which is encouraging. However, it should explicitly clarify that the DL model is trained on outputs from the physics-based model rather than observed data, as its performance reflects replication of the physics-based model rather than independent validation against real-world measurements.

3. Regarding the 2D geomorphological simulation using the physics-based HEC-RAS model, which serves as the reference for the DL model, the physical representation appears relatively simplified, and many key parameters are missing. For example, the Manning's n values, the sediment transport formulations employed (e.g., bedload: van Rijn, 1984; Engelund and Fredsøe, 1976; Meyer-Peter and Müller, 1948) are not specified. It is also unclear how sediment input at the upstream boundary (bedload and/or suspended load) is specified, and how the time steps are configured during the simulation. Furthermore, calibration and validation of the physics-based model appear to be missing. Without these essential steps, it is difficult to assess the reliability of the HEC-RAS model for simulating morphodynamic processes in the study area, which raises concerns about the robustness of the DL model that relies on it as reference data.

4. Similarly, in lines 113–115, the authors state: "*The resulting framework predicts the dynamics of essential hydrodynamic outputs: water depth and flow velocity, which are fundamental inputs to the morphodynamic target, represented in bed change maps.*" and the bed change results appear to rely solely on these two hydrodynamic outputs. This is insufficient for morphodynamic modelling. In addition to hydrodynamic variables, morphodynamic processes also depend on sediment-related

variables, such as sediment concentration (suspended load, which plays an important role in sedimentation within the floodplain), total sediment transport and transport rates (bedload and/or suspended load), and sediment properties (e.g., grain size and settling velocity).

However, I acknowledge that addressing these points (3 and 4) would require considerable time, access to observed data, and expert knowledge in morphodynamic modelling, particularly with a thorough understanding of the study area. Given that the main aim of this study is to demonstrate the ability of the DL model to achieve substantially higher computational efficiency, I leave it to the editor to decide whether these concerns need to be addressed for publication.

5. Data training (lines 144–145): Approximately 11 flood events were used for training, 3 for validation, and 6 for testing, with this information presented only visually in Figure 1b. This approach is rather simple, and it is unclear how this selection was justified. It would therefore be helpful if the authors could include a sentence explaining the rationale for this choice—for example, whether the 11 training events cover the full range of flood magnitudes (from small to extreme events) in terms of probability, total water volume, or peak discharge over the period of record, or at least whether the validation and testing events fall within the range of the training data (quantified). Providing such quantitative information would help readers better understand the representativeness of the training dataset

6. Line 297-300. In physics-based models, error accumulation is a major concern, as small errors at early stages can grow and propagate over time. Here, the CB-trained framework demonstrates rapid recovery after the first testing event and shows no apparent accumulation of relative errors in subsequent events (Fig.6), the reasons for the initially high relative errors are not discussed. Further clarification on the mechanisms underlying this recovery would help readers better understand the robustness and stability of the framework in continuous simulations.

7. Section 3.4: The authors test a range of time steps for the DL model, from 0.25 hr to 16 hr. However, a key detail is missing: the time step used in the physics-based reference model is not provided. This information is important because the physics-based model serves as the baseline for training and evaluating the DL model. For example, if the physics-based model uses a 30-minute or 5-hour time step, it could strongly influence which DL model time step is most appropriate. Once the time step of the physics-based model is provided, it would be useful to discuss the optimal DL time step in relation to the physics-based model, as well as the trade-offs between computational efficiency and prediction accuracy.

8. Lines 420–421 mention Manning's n, grain-size distributions, and cohesive properties, which as come out of nowhere (see point 3).

9. Given the complexity of geomorphological processes, more discussion on the limitations and broader applications of DL is needed, especially since it is purely simulation-based. The DL model is evaluated only against the physics-based HEC-RAS model, not observed field data, meaning any errors in HEC-RAS are inherited. It should be noted that the DL model reflects its ability to replicate HEC-RAS rather than providing independent validation of real-world morphodynamics. The study focuses on a relatively simple domain, a single reach with one boundary condition. It would be valuable for the authors to discuss applying DL models to more complex river systems, such as large-scale river–floodplain networks with multiple boundaries. Additionally, the discussion could address the model's handling of internal morphological changes (e.g., dredging, dam construction, or localized sediment

management) and the lack of explicit incorporation of expert knowledge or physical constraints. Outlining strategies to address these limitations would guide applications of DL models to more realistic, actively managed river systems

**Minor comments:** The use of the Morphological Acceleration Factor (Mf) requires careful consideration as it can affect flood-wave propagation and sediment dynamics. In this study, for example, the duration of discharge exceeding 600 $m^3s^{-1}$ for event Tr-1 is ~10 hours when Mf = 1, but reduces to ~2 hours when Mf = 5 (Figure 1b). With Mf = 1, high discharge persists long enough to propagate downstream and spread onto the floodplain. In contrast, Mf = 5 shortens the high-flow duration, potentially reducing downstream peak discharge due to increased floodplain storage and attenuation. Since sediment transport is strongly non-linear with discharge, these changes in flood-wave dynamics may introduce systematic errors in sediment transport and bed-change simulations. The authors should also highlight the importance of expert knowledge in morphological modelling, not only for physics-based models but also for DL models.

---

## Author Comment (AC1)

EGUSPHERE-2025-3368 | Research article
Title: An Integrated Deep Learning Framework Enables Rapid Spatiotemporal Morphodynamic Predictions Toward Long-Term Simulations
Author(s): Mohamed M. Fathi et al.
MS No.: egusphere-2025-3368
MS type: Research article

The authors appreciate the peer review process facilitated by the Editor at *ESurf*. We extend our sincere thanks to the reviewers for their valuable feedback, which has significantly contributed to the enhancement of this manuscript.

**Reviewer #2**

This paper explores the efficiency and accuracy of using Deep Learning (DL) for predicting of geomorphic (river bed) change, in comparison with using a 2D morphodynamic model. The paper is clearly written and for the most part the figures are understandable and illustrative. I have never seen a study like this before, and I think this is a solid contribution to the literature. It represents a first step into using DL for estimating bed change. The study is relatively limited in scope - at least it seems that way to me. That said, I think this is a great introduction, represents an entire study that others could build on, and it is worthy and ready for publication. For reference, I have personally never used machine learning. I am an experienced numerical modeler but the traditional type. I am excited about the potential of studies like this. However, I am not in a position to review the details on the presentation of HDL-GM and whether "reasonable" choices were made in all cases.

The authors sincerely appreciate the reviewer's time and effort in reviewing this paper. We also value your expertise in identifying missing or unclear points that could have led to misunderstandings, which ultimately enhanced the paper's readability.

The following table contains the authors' responses.

| Comment | Responses (Green) and manuscript modifications (Blue) |
|---|---|
| 1. Given this, I did wonder if there was rationale for the applying HDL-GM to 6 flood events. Was there a reason for 6? There is discussion of how error changes when doing continuous modeling, but was there a reason to stop at 6 events? Is it related to the simulation time required for HEC-RAS? Some discussion of how far forward we can predict would be appreciated by me. Even if we don't know, acknowledging this needs to be studied would be helpful for me. Similarly, I wondered why 11 events were used for training. Is there a relationship between | The authors thank the reviewer for raising this important point regarding the number of events used in the training and testing stages. In general, increasing the size of training datasets may have the potential to enhance the ability of DL models to learn hidden patterns and improve predictive robustness. However, in the context of morphodynamic modeling, the generation of long datasets is constrained by the expensive computational cost of physics-based models to generate training datasets. Furthermore, extending the dataset further would elongate the DL training process, and our main aim was to provide an efficient framework.

In this study, a constructed hydrograph consisting of 20 flood events was selected to span a broad range of flood magnitudes and hydrograph shapes. Repetition of similar events was intentionally avoided, as it is unlikely to provide additional information to the DL framework while substantially increasing computational cost. The partitioning of these 20 events into 11 training, 3 validation, and 6 testing events follows common practice in data-driven modeling, where approximately 50–60% of the data are used for training, with the remainder reserved for validation and independent testing. The six testing events were therefore not |

| | |
|---|---|
| the number of events used for training and the number of events that can be reasonably predicted? | chosen to represent a predictive limit of the framework, but rather to provide a statistically meaningful and computationally feasible basis for evaluating error accumulation and long-term behavior.

The revised manuscript reads as follows:

A hydrograph with a broad range of flood events is essential to enhance the capabilities of the DL framework in capturing hidden patterns in the mapping between inputs and outputs. Unfortunately, generating sufficiently long training and testing datasets is hindered by the long processing time using the physics-based model. Additionally, significant portions of the observed hydrograph are baseflow, which corresponds to a period of relatively low flow with minimal sediment transport. To address these challenges, a constructed hydrograph of 20 events was extracted from the observed hydrograph. This event set was designed to encompass a broader range of flood magnitudes while avoiding long periods of baseflow discharge. Repetition of similar events was intentionally avoided, as such redundancy is unlikely to provide additional informative signals for the DL framework. This hydrograph was split into three portions: training, validation, and testing of 11, 3, and 6 events, respectively (Figure 1-b), following common data-driven modeling practice to balance model learning and independent evaluation within computational constraints.

At present, the relationship between the number of training events and the maximum number of flood events that can be reliably predicted in a continuous morphodynamic framework remains an open research question. This may be constrained by numerous factors, such as the system size, hydrological responses, sediment transport behavior, and the DL architecture itself. We now explicitly acknowledge that the forward prediction horizon requires systematic investigation of longer-term predictive stability in future work. The revised manuscript reads as follows:

These findings highlight the capabilities of the CB-trained framework to predict morphodynamic behavior across a testing dataset encompassing a series of continuous events. Nevertheless, while the HDL-GM framework remains stable, the maximum forward prediction horizon over which robust morphodynamic predictions can be maintained requires further investigation, beyond the scope of this study, and is expected to depend on numerous factors such as hydrologic forcing, geomorphic nonlinearity, and the inductive biases of the DL architecture itself. |
| 2. Another thing I wondered was whether you could use a shorter reach for training but make predictions on a longer reach of river than was used for training. Is that fair? It could be a way to save some training time. Is this a no-no in geomorphology machine learning world? | This is an excellent and timely question. In the present study, training and prediction were performed on the same spatial domain. In fact, this is a critical limitation for such DL algorithms. We acknowledged this clearly in the conclusion section as follows:

Despite these promising capabilities, its primary limitation lies in its inability to effectively simulate unseen topographical conditions. Additionally, the current implementation of the HDL-GM framework is for a particular set of hydrodynamic and sediment transport properties, especially Manning n, grain-size distributions, and cohesive properties. To address these limitations, future work is needed to improve the generalizability and performance by integrating essential hydrodynamic and sediment transport physics into the DL framework (Karniadakis et al., 2021; Mohamad et al., 2021). |

| | Recent advances in temporal DL have enabled models to be trained simultaneously across multiple locations and to generalize predictions to ungauged or unseen sites. Nevertheless, these approaches are predominantly limited to temporal representations and have not yet been extended to fully spatiotemporal modeling frameworks. The field is evolving rapidly, and ongoing efforts, both by our team and in the broader community, are focused on addressing these regionalization challenges. |
|---|---|
| 3. The authors do discuss caveats and limitations, so it's not that those are missing. The points in the two above paragraphs were ones that came up for me as I read, but if the authors feel that addressing these is not useful, I defer to them. | The authors appreciate the reviewer's comment. We have acknowledged this limitation, particularly given that this study is among the first to employ spatiotemporal deep learning frameworks for morphodynamic simulations. We agree that the community is actively working toward more efficient models with improved generalizability to unseen scenarios. |
| 4. L 134,135: What impact does the trapezoidal cross-section assumption make? I did not see a mention of river depth measurements, but wouldn't those be needed to create the cross-sections? | The authors thank the reviewer for highlighting this missing point. The trapezoidal cross-section was introduced due to the absence of bathymetric data in the DEM, following field measurements reported by Costigan et al. (2014).

 The revised manuscript reads as follows:

 Due to the absence of bathymetry information in the DEM file, a trapezoidal cross-section profile was burned into the DEM file (Choné et al., 2018), using a channel depth of 1.1 m and a bankfull width of 95 m, informed by field measurements reported by Costigan et al. (2014). |
| 5. Figure 2C: This is a little confusing to me. In C, isn't it the topography at time t-1 that informs the water depth at time t? Based on this diagram, there is just a continual loop at time t. I think, that in the loop from slice 4 to 1, there is an increase in t. Maybe you could show that in the arrow between slice 4 and slice 1? | The reviewer is correct, and we appreciate this careful reading. The framework operates through an integrated four-step loop: (1) hydrodynamic variables are predicted using inputs from topography time t-1; (2) these hydrodynamic predictions are then used to estimate bed change at time t-1; (3) the predicted bed change is applied to update the topography to time t; and (4) the updated topography is subsequently used as the primary input for the next prediction step. The original version of Figure 2C did not sufficiently emphasize the increment in time between successive iterations, which may have led to confusion.

 The revised figure is as follows: |

**c) Models' integration**

| | |
|---|---|
| 6. L 225-227: I think the reference dataset is from HEC-RAS and the simulated is from HDL-GM? Maybe you could say that directly, or clarify what they are if I am wrong. | The reviewer is correct in their interpretation: in our terminology, the simulated results correspond to the DL model outputs, whereas the reference denotes the physics-based model results. The revised manuscript now clarifies this distinction explicitly in the relevant sections. |
| | To evaluate the accuracy of the proposed model in predicting the hydrodynamic and bed change variables, Root Mean Square Error (RMSE) is used to quantify the average magnitude of errors between HDL-GM predictions and reference results obtained from the HEC-RAS model |
| 7. Equations 1 - 3: I think R and S are bed elevation change. Can you say that explicitly? | Agreed. We planned to use those performance criteria to evaluate the models' performance at both hydrodynamic and morphodynamic levels. That's why we didn't clearly state that those are bed change criteria. However, we agree that explaining this clearly enhances the readability of the paper. |
| | where $R_i, S_i$ are the physics-based reference values obtained from HEC-RAS and the corresponding simulated values predicted by the HDL-GM framework, respectively, and $\bar{R}$ is the average value of reference grids. $N$ is the number of active cells within the predicted or reference domains. These performance criteria are applicable to both the hydrodynamic components (water depth and flow velocity) and the morphodynamic component (bed elevation change). |
| 8. Paragraph around line 245, about the EB testing: For the EB scenario, does every event start on the same initial topography? That is event 1 and event 2 are processed in parallel, but starting on the same topography? | Yes. In the EB framework, each flood event is simulated independently and initialized from the same baseline topography. The authors have revised the manuscript to enhance readability and to present this experiment more clearly for all readers. |
| | Two distinct versions of the HDL-GM framework were tested. These versions are based on the nature of the training dataset: Event-Based (EB) and Continuous-Based (CB) datasets (Figure 1-b). The EB dataset aggregates discrete events that are simulated independently using the physics-based model, each initialized from the same topographic state. This formulation |

| | enhances computational efficiency in generating the training dataset by allowing parallel simulations. Conversely, the CB dataset is derived from a single temporally continuous simulation that spans a sequence of events, which demands significantly longer processing time due to its continuous simulation to resolve the full temporal evolution of the system. |
|---|---|
| 9. Figure 3: Is absolute error the absolute value of R-S at a location? And the relative error is relative to what? I don't see equations for these values. Couldn't relative error exceed 100%? Is the color bar saturated at 100% but values go beyond 100%? | Yes. Absolute error is defined as \|R–S\|. Relative error is computed as the absolute error divided by the reference error at each grid cell. As noted by the reviewer, the relative error can exceed 100%. For visualization purposes, the color bar is intentionally saturated at 100% to improve figure clarity, as values above this threshold uniformly indicate very poor agreement regardless of their exact magnitude. We also added a full paragraph in the methodology section to define how to compute both absolute error and relative error.

Although NRMSE and R² provide useful global measures of model performance, geomorphic responses are spatially heterogeneous and cannot be fully described by a single performance value. To better capture spatial error patterns, cell-scale absolute and relative error maps were developed, comparing the physics-based HEC-RAS reference results with the HDL-GM predictions. Absolute error represents the absolute difference between the reference and predicted values at cell-scale, with a perfect value of zero and no upper bound. Relative error expresses this absolute error normalized by the reference value at each cell, enabling comparison across regions with differing response magnitudes. While relative error also attains an optimal value of zero, it may exceed 100% in areas where predictions are large relative to the reference bed change values or the reference bed change values are marginal. Performance criteria along spatial error metrics provide a more complete assessment of both the magnitude and significance of prediction errors across the domain. |
| 10. L 312: "95% error criterion indicates that 95% of the active cells exhibit an error below this value." What is "this value"? | The authors appreciate the reviewer's comment. We agree that the original phrasing was unclear. We revised it as follows:

The 95% error is defined as the threshold below which 95% of the cell-scale absolute errors across all active cells fall, corresponding to the 95th percentile of the absolute error distribution. Consequently, it offers additional insight beyond mean-based statistics by quantifying the magnitude of errors affecting the vast majority of active cells, defined here as cells exhibiting bed changes greater than 0.02 m. |

---

## Author Comment (AC2)

EGUSPHERE-2025-3368 | Research article
Title: An Integrated Deep Learning Framework Enables Rapid Spatiotemporal Morphodynamic Predictions Toward Long-Term Simulations
Author(s): Mohamed M. Fathi et al.
MS No.: egusphere-2025-3368
MS type: Research article

The authors appreciate the peer review process facilitated by the Editor at *ESurf*. We extend our sincere thanks to the reviewers for their valuable feedback, which has significantly contributed to the enhancement of this manuscript.

**Reviewer #1**

I was invited to review the manuscript by Mohamed et al. (2025), which presents an integrated deep learning (DL) framework trained on outputs from a physics-based morphodynamic model to efficiently accelerate spatiotemporal predictions of river–floodplain evolution. While the framework does not aim to introduce new physical concepts or process understanding, it effectively leverages deep learning to reproduce the behavior of the reference physics-based model in a computationally efficient manner. Consequently, the primary contribution of the manuscript lies in methodological advancement and workflow efficiency rather than in the development of new morphodynamic theory. Overall, the study is interesting and promising. Nevertheless, several points would benefit from clarification or further refinement. Addressing these comments to the satisfaction of the editor and reviewers would, in my view, strengthen the manuscript, and I would be happy to recommend it for publication.

The authors sincerely appreciate the reviewer's time and effort in reviewing this paper. We value your expertise and thank you for the insightful suggestions, which have helped us clarify key methodological details and improve the overall clarity of the paper.

The following table contains the authors' responses.

| Comment | Responses (Green) and manuscript modifications (Blue) |
|---|---|
| 1. The main focus of this study is to demonstrate that deep learning (DL), trained on outputs from the physics-based model, can reproduce comparable results at a substantially higher computational speed. It does not explore the broader potential of DL for simulating hydrodynamic and morphodynamic processes in river or floodplain systems. Therefore, the main contribution lies in computational efficiency rather than providing new physical insights. In addition, numerous studies have applied deep learning to achieve higher computational speeds; for example, Synthetic in BenTIvoglio et al. (2022, *Deep learning methods for flood mapping: a review of existing applications and future research directions*) | The authors appreciate the reviewer for this thoughtful comment. We agree that computational efficiency is a central contribution of this study, specifically achieved through a spatiotemporal hybrid CNN–LSTM framework that jointly predicts hydrodynamic variables (water depth and flow velocity) and 2D morphodynamic responses (bed-change maps) on structured grids.

While numerous studies have leveraged DL to develop hydrodynamic surrogates, especially flood mapping (such as the papers mentioned by the reviewer: Bentivoglio et al., 2022; Karim et al., 2023), we are not aware of prior work demonstrating an end-to-end, 2D morphodynamic map prediction framework that explicitly predicts bed-level change within an updated system state iteratively over multiple flood events to assess long-horizon stability. In this sense, the novelty lies not only in acceleration, but also in the integrated DL framework to deliver hydrodynamic and morphodynamic predictions at the spatiotemporal scale. |

| | |
|---|---|
| and Karim et al. (2023, *A review of hydrodynamic and machine learning approaches for flood inundation modeling*). Consequently, the statement in the abstract (line 34) claiming that this study represents "pioneering new frontiers in fluvial morphodynamic modelling" is misleading and should be revised. | However, we have revised the statement according to the reviewer's comment, as follows:

This novel approach generates predictions 4700 times faster than traditional physics-based computational models, representing a paradigm shift in long-term river evolution simulations and opening new opportunities for fluvial morphodynamic modeling. |
| 2. The abstract reports good performance metrics (e.g., RMSE, R²), which is encouraging. However, it should explicitly clarify that the DL model is trained on outputs from the physics-based model rather than observed data, as its performance reflects replication of the physics-based model rather than independent validation against real-world measurements. | The authors thank the reviewer for this comment. We agree on this. The revised sentence reads as follows:

This paper introduces and tests a robust Deep Learning (DL) framework that opens the door to overcoming these challenges through integrating convolutional neural networks (CNNs) with long short-term memory algorithms (LSTM) architectures, trained using outputs from the physics-based HEC-RAS model. |
| 3. Regarding the 2D geomorphological simulation using the physics-based HEC-RAS model, which serves as the reference for the DL model, the physical representation appears relatively simplified, and many key parameters are missing. For example, the Manning's n values, the sediment transport formulae employed (e.g., bedload: van Rijn, 1984; Engelund and Fredsøe, 1976; Meyer-Peter and Müller, 1948) are not specified. It is also unclear how sediment input at the upstream boundary (bedload and/or suspended load) is specified, and how the time steps are configured during the simulation. Furthermore, calibration and validation of the physics-based model appear to be missing. Without these essential steps, it is difficult to assess the reliability of the HEC-RAS model for simulating morphodynamic processes in the study area, which raises concerns about the robustness of the DL model that relies on it as reference data. | The authors thank the reviewer for this detailed and important comment. We agree that, for studies aiming at predicting morphodynamic behavior, extensive parameter specification, calibration, and validation of the physics-based model are essential. However, the primary objective of this study is fundamentally different. Here, we aim to assess the capability of a DL framework to learn, reproduce, and efficiently emulate the spatiotemporal behavior of a physics-based morphodynamic model, rather than to optimize agreement with field observations, which are unavailable for the Ninnescah River. Accordingly, the HEC-RAS model is employed as a synthetic reference generator that produces internally consistent morphodynamic dynamics. This simulator-based training strategy is well established in surrogate and emulator modeling and aligns with several recent papers, such as "Next-generation deep learning based on simulators and synthetic data" by De Melo et al. (2022) who highlighted that synthetic data offer a practical solution to data scarcity by providing abundant, noise-free, and fully annotated training samples, thereby enabling effective DL training while avoiding many of the logistical challenges associated with observational datasets.

Due to the limited availability and spatial coverage of observational data in the study reach, calibration and validation of parameters such as Manning's roughness coefficient and sediment transport parameters were not pursued. Instead, we adopted a consistent model configuration to isolate the DL model's ability to capture nonlinear morphodynamic responses governed by the physics-based solver. Two Manning's roughness coefficient of 0.035 and 0.05 were used within the river reach and surrounding floodplain, respectively, which falls well within the commonly accepted range for natural alluvial channels and floodplains.

The authors agreed that the Methodology section would benefit from more information about the hydrodynamic and morphodynamic setting used to produce the training dataset. We also wanted to highlight that the reported DL performance reflects its capacity to reproduce the morphodynamic |

responses generated from the physics-based model, not its ability to predict real-world morphodynamics directly.

We have clarified this distinction explicitly in the revised manuscript across different locations:

Methodology: we added a new paragraph

Due to the limited availability of spatiotemporal observational data within the study reach, full calibration and independent validation of the hydraulic and sediment transport parameters were not feasible. To maintain consistency in the physics-based reference, we adopted a physically reasonable and internally coherent HEC-RAS configuration. Uniform Manning's n values of 0.035 for the main channel and 0.05 across the floodplain were applied, which fall within commonly reported ranges for alluvial systems. HEC-RAS includes a wide suite of empirical sediment-transport formulations, and for this study, the Wu equation was selected due to its demonstrated robustness in comparative assessments against other widely used transport formulas (Wu et al., 2000). The Wu equation is well-suited for channels with the grain size distribution of the Ninnescah River, as it explicitly incorporates the effects of nonuniform sediment mixtures and gradation (Wu and Lin, 2014). In contrast, classical approaches such as Meyer-Peter–Müller or Engelund–Hansen do not incorporate bed-material nonuniformity and may therefore yield biased transport estimates under conditions similar to those of the study reach (Hunziker and Jaeggi, 2002). The developed model ensures that the morphodynamic responses arise from physics-based background, thereby allowing a more direct evaluation of the DL model's ability to replicate the nonlinear morphodynamic behavior of the physics-based solver.

Conclusion: we added a new paragraph

This study introduces a novel DL framework designed to address the computational challenges of physics-based models, enabling robust, rapid, and reliable generation of 2D morphodynamic maps. Given the lack of observational data for the Ninnescah River, this study does not aim to optimize agreement with field measurements; instead, it evaluates the capacity of the DL model to reproduce the spatiotemporal dynamics generated by a physics-based numerical solver. For this purpose, the HEC RAS model was used as a synthetic reference generator, providing internally consistent hydrodynamic and morphodynamic responses. This learning approach, widely used in surrogate modeling and increasingly adopted in Earth-surface research where observational datasets are sparse, leverages a synthetic, noise-free, and complete dataset to enable robust DL training while avoiding the practical constraints of field data collection (De Melo et al., 2022).

| 4. Similarly, in lines 113–115, the authors state: "*The resulting framework predicts the dynamics of essential hydrodynamic outputs: water depth and flow velocity, which are fundamental inputs to the morphodynamic target, represented in bed change* | The authors thank the reviewer for this insightful comment and fully agree that classical physics-based morphodynamic models explicitly rely on sediment-related variables such as suspended load, bedload transport rates, and sediment properties, which govern bed evolution processes. However, the role of these variables differs fundamentally between physics-based solvers and data-driven emulation frameworks. The objective of this study is to emulate the input–output behavior of a physics-based morphodynamic model. In this context, the proposed framework is designed to learn a direct mapping |

*maps.*" and the bed change results appear to rely solely on these two hydrodynamic outputs. This is insufficient for morphodynamic modelling. In addition to hydrodynamic variables, morphodynamic processes also depend on sediment-related variables, such as sediment concentration (suspended load, which plays an important role in sedimentation within the floodplain), total sediment transport and transport rates (bedload and/or suspended load), and sediment properties (e.g., grain size and settling velocity).

However, I acknowledge that addressing these points (3 and 4) would require considerable time, access to observed data, and expert knowledge in morphodynamic modelling, particularly with a thorough understanding of the study area. Given that the main aim of this study is to demonstrate the ability of the DL model to achieve substantially higher computational efficiency, I leave it to the editor to decide whether these concerns need to be addressed for publication.
* * *
between hydrodynamic forcings and resulting bed elevation changes as produced by the HEC-RAS solver, without requiring explicit representation of intermediate sediment transport variables.

By conditioning the DL framework on key hydrodynamic variables such as water depth and flow velocity, which strongly control sediment mobilization and transport capacity, the model is able to learn statistical relationships that implicitly reflect the influence of sediment-related processes. In practice, variables such as suspended sediment concentration and transport rates are often strongly correlated with flow depth, velocity, and hydrograph characteristics, allowing their effects to be captured indirectly through these hydrodynamic predictors.

We have revised the manuscript to clarify this point in the Methodology Section as follows:

To maximize the efficiency of the HDL framework as a physics-based emulator, the model was designed to learn the final morphodynamic response produced by HEC-RAS rather than to replicate intermediate sediment-transport processes. Accordingly, the HDL architecture was trained to predict bed-level change directly, using hydrodynamic fields (water depth and flow velocity) together with the upstream hydrograph as inputs. This reduced-input configuration was intentionally adopted to evaluate the model's capacity to infer the morphodynamic patterns from the essential driving variables, while avoiding additional complexity associated with explicitly modeling suspended sediment, bedload transport rates, or sediment-property fields.
* * *
5. Data training (lines 144–145): Approximately 11 flood events were used for training, 3 for validation, and 6 for testing, with this information presented only visually in Figure 1b. This approach is rather simple, and it is unclear how this selection was justified. It would therefore be helpful if the authors could include a sentence explaining the rationale for this choice—for example, whether the 11 training events cover the full range of flood magnitudes (from small to extreme events) in terms of probability, total water volume, or peak discharge over the period of record, or at least whether the validation and testing events fall within the range of the training data (quantified). Providing such quantitative information would help readers better understand the representativeness of the training dataset
* * *
The authors thank the reviewer for raising this important point regarding the number of events used in the training and testing stages. This comment was repeated by the second reviewer, and we agree that it wasn't clear enough in the first version.

In general, increasing the size of training datasets may have the potential to enhance the ability of DL models to learn hidden patterns and improve predictive robustness. However, in the context of morphodynamic modeling, the generation of long datasets is constrained by the expensive computational cost of physics-based models to generate training datasets. Furthermore, extending the dataset further would elongate the DL training process, and our main aim was to provide an efficient framework.

In this study, a constructed hydrograph consisting of 20 flood events was selected to span a broad range of flood magnitudes and hydrograph shapes. Repetition of similar events was intentionally avoided, as it is unlikely to provide additional information to the DL framework while substantially increasing computational cost. The partitioning of these 20 events into 11 training, 3 validation, and 6 testing events follows common practice in data-driven modeling, where approximately 50–60% of the data are used for training, with the remainder reserved for validation and independent testing. The six testing events were therefore not chosen to represent a predictive

limit of the framework, but rather to provide a statistically meaningful and computationally feasible basis for evaluating error accumulation and long-term behavior.

The revised manuscript reads as follows:

Additionally, significant portions of the observed hydrograph are baseflow, which corresponds to a period of relatively low flow with minimal sediment transport. To address these challenges, a constructed hydrograph of 20 events was extracted from the observed hydrograph. This event set was designed to encompass a broader range of flood magnitudes while avoiding long periods of baseflow discharge. Repetition of similar events was intentionally avoided, as such redundancy is unlikely to provide additional informative signals for the DL framework. This hydrograph was split into three portions: training, validation, and testing of 11, 3, and 6 events, respectively (Figure 1-b), following common data-driven modeling practice to balance model learning and independent evaluation within computational constraints.

| 6. Line 297-300. In physics-based models, error accumulation is a major concern, as small errors at early stages can grow and propagate over time. Here, the CB-trained framework demonstrates rapid recovery after the first testing event and shows no apparent accumulation of relative errors in subsequent events (Fig.6). The reasons for the initially high relative errors are not discussed. Further clarification on the mechanisms underlying this recovery would help readers better understand the robustness and stability of the framework in continuous simulations. | The authors appreciate the reviewer's comment. We agree that this point requires explanation. The revised manuscript reads as follows:

Figure 5 presents the absolute and relative error results at the end of multiple testing events within the CB dataset. Initially, the CB-trained framework maintained relatively high relative errors at the end of the first testing event, primarily because the reference bed change magnitudes during this first event are small; as a result, even minor absolute deviations from the HDL-GM predictions translate into large relative errors. After this first event, it exhibited rapid recovery, yielding substantially reduced relative errors from the second testing event. This robust performance continued throughout the CB testing dataset, with no accumulation of relative errors. |
| --- | --- |
| 7. Section 3.4: The authors test a range of time steps for the DL model, from 0.25 hr to 16 hr. However, a key detail is missing: the Time step used in the physics-based reference model is not provided. This information is important because the physics-based model serves as the baseline for training and evaluating the DL model. For example, if the physics-based model uses a 30-minute or 5-hour time step, it could strongly influence which DL model time step is most appropriate. Once the time step of the physics-based model is provided, it would be useful to discuss the optimal DL time step in relation to the physics- | The authors thank the reviewer for this important point. We agree that the reference solver's time stepping should be reported because it allows the comparison between traditional physics-based and DL models.

Physics-based morphodynamic solvers such as HEC-RAS require very small computational time steps to maintain numerical stability in both the hydrodynamic and sediment-transport components. In our configuration, the HEC-RAS hydrodynamic time step was 3 seconds, which is a primary reason for the long processing times. One advantage of the proposed DL framework is that, unlike the physics-based solver, it does not inherit CFL-based stability constraints or sediment-transport time-step limitations. This flexibility allows the DL model to operate at much coarser temporal resolutions (hours) while still capturing the essential morphodynamic evolution. The temporal-resolution analysis in Section 3.4 was therefore designed to evaluate how coarse the DL time step can be made before losing fidelity to the physics-based baseline. |

| | |
|---|---|
| based model, as well as the trade-offs between computational efficiency and prediction accuracy. | The revised manuscript reads as follows:

The prediction runtime and computational demands have been critical considerations in the morphodynamic modeling approach. This section highlights the substantial efficiency gains achieved by the proposed HDL-GM framework relative to the HEC-RAS simulations. Notably, HEC-RAS requires a 3-second computational time step to maintain numerical stability in both hydrodynamic and sediment-transport solvers. This constraint is a major source of the long processing times associated with physics-based models. In contrast, the HDL-GM framework is not limited by such stability limitations and can therefore operate at much coarser time steps, allowing evaluations at coarser resolutions, enabling the substantial computational speedup demonstrated by the DL-based approach. |
| 8. Lines 420–421 mention Manning's n, grain-size distributions, and cohesive properties, which as come out of nowhere (see point 3). | The authors agree with the reviewer's comment, and we revised the manuscript and provided more information about the modeling part and the utilized Manning n value.

Uniform Manning's n values of 0.035 for the main channel and 0.05 across the floodplain were applied, which fall within commonly reported ranges for alluvial systems. |
| 9. Given the complexity of geomorphological processes, more discussion on the limitations and broader applications of DL is needed, especially since it is purely simulation-based. The DL model is evaluated only against the physics-based HEC-RAS model, not observed field data, meaning any errors in HEC-RAS are inherited. It should be noted that the DL model reflects its ability to replicate HEC-RAS rather than providing independent validation of real-world morphodynamics. The study focuses on a relatively simple domain, a single reach with one boundary condition. It would be valuable for the authors to discuss applying DL models to more complex river systems, such as large-scale river– floodplain networks with multiple boundaries. Additionally, the discussion could address the model's handling of internal morphological changes (e.g., dredging, dam construction, or localized sediment management) and the lack of explicit incorporation of expert knowledge or physical constraints. Outlining strategies to address these limitations would guide applications of DL models to more realistic, actively managed river systems | The authors thank the reviewer for this thoughtful comment and agree that a clear discussion of limitations and broader applicability is essential. We clearly highlighted that the DL framework presented here is evaluated exclusively against the physics-based HEC-RAS model, and therefore its performance reflects the ability to replicate the numerical solver rather than to provide validation of observed morphodynamic behavior.

Abstract

This paper introduces and tests a robust Deep Learning (DL) framework that opens the door to overcoming these challenges through integrating convolutional neural networks (CNNs) with long short-term memory algorithms (LSTM) architectures, trained using outputs from the physics-based HEC-RAS model.

Conclusion

Given the lack of observational data for the Ninnescah River, this study does not aim to optimize agreement with field measurements; instead, it evaluates the capacity of the DL model to reproduce the spatiotemporal dynamics generated by a physics-based numerical solver. For this purpose, the HEC-RAS model was used as a synthetic reference generator, providing internally consistent hydrodynamic and morphodynamic responses.

We also agree that extending the framework to more complex river–floodplain systems represents an important direction for future work. We have expanded the conclusion to explicitly outline these limitations and potential strategies for applying DL-based emulators to larger and more complex systems. |

| | |
|---|---|
| | **Conclusion**

Ultimately, the proposed framework demonstrates significant potential in enhancing morphodynamic modeling in fluvial rivers and floodplains, particularly from the long-term perspective, climate change assessments, river rehabilitation projects, and other investigations requiring long-term simulations. Despite these promising capabilities, its primary limitation lies in its inability to effectively simulate unseen topographical conditions. In addition, the current application is limited to a single reach with one upstream boundary condition; future work should therefore evaluate the framework's performance in larger and more complex river–floodplain networks with multiple boundary conditions. The current implementation is also tailored to a particular set of hydrodynamic and sediment transport properties, especially Manning n, grain-size distributions, and cohesive properties, which may limit generalizability. Addressing these constraints will require integrating essential hydrodynamic and sediment transport physics into the DL framework (Karniadakis et al., 2021; Mohamad et al., 2021). |
| **Minor comments:** The use of the Morphological Acceleration Factor (Mf) requires careful consideration as it can affect flood-wave propagation and sediment dynamics. In this study, for example, the duration of discharge exceeding 600 m³s-1 for event Tr-1 is ~10 hours when Mf = 1, but reduces to ~2 hours when Mf = 5 (Figure 1b). With Mf = 1, high discharge persists long enough to propagate downstream and spread onto the floodplain. In contrast, Mf = 5 shortens the high-flow duration, potentially reducing downstream peak discharge due to increased floodplain storage and attenuation. Since sediment transport is strongly non-linear with discharge, these changes in floodwave dynamics may introduce systematic errors in sediment transport and bed-change simulations. The authors should also highlight the importance of expert knowledge in morphological modelling, not only for physics-based models but also for DL models. | The authors appreciate this important point. We agree that the initial version of the manuscript did not sufficiently explain the rationale for selecting an acceleration factor of 5. In the revised manuscript, we have expanded this section to clarify both why this value was chosen and how higher values of Mf can adversely affect morphodynamic accuracy. The updated manuscript reads as follows:

It is worth noting that there is a trade-off between utilizing a higher acceleration factor and the accuracy of the morphodynamic outputs. Morgan et al. (2020) explored this trade-off on Nooksack River in Washington through investigating a range of Mf from 5 to 50. Their results showed that Mf of 5 provided a valuable balance, reducing computational time by approximately 80% while maintaining a relatively low absolute percentage error of about 8%. In contrast, larger acceleration factors (Mf over 20) led to substantial increases in error, in some cases exceeding 30%, indicating degradation of morphodynamic fidelity. Because such nonlinearities affect both the hydrograph and the morphological responses, expert judgment remains essential for interpreting morphodynamic outputs. Based on these considerations, Mf of 5 was adopted in the present study, where each flood event spans a full-scale of 10-days, encompassing the typical progression of flood events from initial baseflow, through the peak, and returning to baseflow, which is reduced to only 2 days using Mf of 5 (Figure 1-b). |